# Analysis of Anxiety, Depression and Aggression in Patients Attending Pain Clinics

**DOI:** 10.3390/ijerph15122898

**Published:** 2018-12-18

**Authors:** Dariusz Kosson, Małgorzata Malec-Milewska, Robert Gałązkowski, Patryk Rzońca

**Affiliations:** 1Department of Anaesthesiology and Intensive Care, Medical University of Warsaw, Division of Teaching, 4 Lindley St., 02-005 Warsaw, Poland; kosson@wp.pl; 2Department of Anesthesiology and Intensive Care, Medical Center for Postgraduate Education, 231 Czerniakowska St., 00-416 Warsaw, Poland; lmilewski@post.pl; 3Department of Emergency Medical Services, Faculty of Health Science, Medical University of Warsaw, 81 Żwirki i Wigury St., 02-091 Warsaw, Poland; r.galazkowski@lpr.com.pl; 4Department of Emergency Medicine, Faculty of Health Sciences, Medical University of Lublin, 4-6 Staszica St., 20-081 Lublin, Poland

**Keywords:** pain, anxiety, depression, aggression, pain clinics

## Abstract

The aim of the study was to measure the frequency of such emotional disturbances as anxiety, depression and aggression among patients treated in a pain clinic, as well as assess the factors contributing to such disorders. Research was conducted from January 2014 to April 2018 and involved patients treated in two pain clinics in the city of Warsaw, Poland. The study used the Hospital Anxiety and Depression Scale—Modified Version (HADS-M) and the Numerical Rating Scale (NSR). 1025 patients were recruited. The main reasons for their attending the pain clinic were osteoarticular pain (43.61%) and neuropathic pain (41.56%). Emotional disturbances in the form of anxiety were diagnosed in 32.39% of all the patients, depression in 17.85%, and aggression in 46.15%. The factors determining the level of anxiety in the study group were: sex, age, pain intensity and the lack of pharmacological treatment. Depression was determined by sex, pain intensity and the time of treatment in the clinic, while aggression by age and pain intensity.

## 1. Introduction

Pain can be either nociceptive or non-nociceptive, usually neuropathic, acute or chronic. It is mostly defined and perceived as an unpleasant sensory and emotional experience associated with actual or potential tissue damage or described in terms of such damage [1]. For each individual it is a unique sensory experience, characterized by location, duration and type but also depending on the personal experiences of each person. This is why pain is one of the main reasons why patients report at healthcare facilities. Studying this phenomenon requires considering its physical, psychosocial, as well as spiritual aspects. Experiencing pain has an impact on the life and functioning of the individuals involved, resulting in the worsening of their own quality of life but also that of the people close to them, thus leading to ensuing social consequences [1,2,3,4,5,6]. Additionally, in the long term, experiencing pain can contribute to the occurrence of emotional problems, or mood alterations in the form of anxiety or depression, resulting in a significant limitation of the person’s functioning in social life. [4,7,8].

Pain-related mental changes lead to many negative symptoms associated with cognitive, emotional and behavioral disorders. In most patients, emotional stress takes the form of depression, anxiety and anger. Daily activities frequently become dominated by anxiety and fears associated with the progress of pain symptoms. It must be underlined that negative emotions associated with pain are often suppressed. Thus, aggression fueled by anger is often repressed, resulting in the accumulation of negative emotions and consequently depression. Depression in connection with anxiety is considered an important factor in the pathogenesis of somatization. [4,7,8].

Chronic pain is a little-known but increasingly prevalent condition posing a major diagnostic challenge, which makes it difficult to choose appropriate treatment. On the other hand, significant progress can be observed in the treatment of acute pain in such cases as: injury, myocardial infarction, childbirth, palliative care and neoplasms. This is due to the growing knowledge about available, appropriate pain management medication, its dosage and the way it should be administered. Nevertheless, in spite of this advancement in understanding pain, many attempts still fail to relieve it and improve the quality of the patients’ everyday life. An important element in the effective treatment of pain, especially of the kind not responding to the classic first-line drugs, is to include biological, psychological and social aspects in managing both acute and chronic pain [5,9,10,11,12].

Moreover, gaining thorough knowledge of the symptoms experienced—not only those of a physical nature—as well as taking into account the expectations of the patient, makes it possible to develop comprehensive plans of the pain management process. Due to the complexity of such procedures, many countries have set up specialized organizational units to implement comprehensive, interdisciplinary treatment of patients suffering from pain and discomfort. Examples of this sort can be found in pain clinics in Germany and in Poland [5,9,13].

It is for this reason that the authors of the present paper set as their objective to measure the frequency of emotional disturbances in the form of anxiety, depression and aggression among those treated in pain clinics and to specify the factors that influence such states.

## 2. Materials and Methods

### 2.1. Participants

Research was conducted from January 2014 to April 2018 in patients under the care of two out-patient pain clinics in the city of Warsaw: The Pain, Anesthesiology and Intensive Therapy Clinic of the Medical University of Warsaw, 4 Lindleya Street, as well as the Pain Clinic located at 231 Czerniakowska Street (Poland). The inclusion criteria for participation in the study were: 18 years of age, ascertainment of pain on the basis of the patient’s medical history, and attending a pain clinic in the city of Warsaw. The exclusion criterion was lack of informed consent on the part of the patient. The study was conducted in accordance with the Helsinki Declaration and obtained approval no. AKBE/24/15 from the Bioethics Committee of the Medical University of Warsaw. It was, moreover, approved by the managers of the clinics involved. The patients taking part in the project were informed about the voluntary nature of their participation, their anonymity, and the fact that the results of the study would be used exclusively for scientific purposes.

### 2.2. Assessment

Research was based on the analysis of the medical histories of the patients attending the pain clinic (their sociodemographic data) as well as by the diagnostic survey method using the questionnaire technique. The research tools were: the Hospital Anxiety and Depression Scale—Modified Version (HADS-M) and the NSR—Numerical Rating Scale.

The Hospital Anxiety and Depression Scale—Modified Version (HADS-M) is a revision of the Hospital Anxiety and Depression Scale (HADS) developed by Zigmond and Snaith [9], and was adapted for Polish conditions by Majkowicz, de Walden-Gałuszko and Chojnacka-Szawłowska. HADS-M comprises three subscales to assess anxiety, depression and aggression. The subscale of both anxiety and depression consists of seven statements, while the subscale of aggression two. Answers are given on a scale of 0 to 3 points. The maximum score for the anxiety and depression subscale is 21, while for the aggression subscale, it is 6 points. In total the person examined can score a maximum of 48 points. Higher results on each subscale point to greater disturbances. Anxiety, depression and aggression are assessed and interpreted separately using the percentage reference to the score (lack of disturbances: 0–33.33%; borderline cases: 33.34–47.62%; the occurrence of threat: 47.63–100%) or using the score matrix: lack of disturbances: 0–7 points for anxiety and depression, 0–2 points for aggression; borderline cases: 8–10 points for anxiety and depression, 3 points for aggression; disturbances occurring: 11–21 points for anxiety and depression, 4–6 points for aggression. The validation analysis of the basic and modified version of the HADS scale showed satisfactory reliability and accuracy. Spearman’s rank correlation coefficient between the scale positions and the overall result of the given subscale was statistically significant (*p* ≥ 0.01) and within the range of 0.41–0.76 [14,15,16].

The Numerical Rating Scale—NRS—is a one-dimensional measurement of pain intensity in adults. The most frequently used version is the 11-point NRS scale, which was the one used in the present study. The NRS form can be filled out by reporting scores verbally or by the patients themselves in a graphic way. The respondents are asked to rate their pain along the line that best represents the intensity of their pain. The NRS score ranges from 0 to 10, with 0 indicating no pain and 10 the worst imaginable pain. Higher scores on the NRS scale indicate a higher intensity of pain [17,18].

### 2.3. Statistical Analysis

The results were statistically analyzed using the STATISTICA version 12,5 software (StatSoft, Kraków, Poland). Amount and percentages were used in the description of quality data, while in numerical data—the mean (M) and Standard Deviation (SD).

The normality of variable distribution was performed using the Shapiro-Wilk normality test. Regression analysis using the enter method was performed in order to assess the impact of each variable on the particular subscales in HADS-M: anxiety, depression and aggression. Data interpretation is based on comparing the value of the β coefficient, which is interpreted in the categories of direction and strength depending on particular predictors. Owing to this, it is possible to say which variable has a greater impact on the dependent variable. The degree of the regression model fitting the data is the goodness-of-fit value, R^2^—the higher it is the better the fit of the model. The study assumed statistical significance as *p* < 0.05.

## 3. Results

The study enrolled 1025 people, mostly women (63.71%). Their age ranged from 65–79 (35.22%), on average 62.15 (±15.48). The main reasons for attending the pain clinic were: osteoarticular pain (43.61%) and neuropathic pain (41.56%). Most patients were given specialist treatment in accordance with the principles of multimodal therapy using pharmacological treatment that includes non-analgesic medication (54.43%), with antidepressant drugs being most frequent. For the majority this was not their first visit (75.02%), and the time of treatment was usually from 2 to 5 years (24.45%). The mean pain level on the NRS scale was 4.30 (±2.87). 

Table 1 presents the characteristics of the participants in the study. Anxiety was diagnosed in one third of the patients (32.39%), depression occurred in almost one fifth (17.85%), while aggression in almost half of the respondents (46.15%) (Table 2).

Table 3 shows regression analysis on the subscale of HADS-M Anxiety. The regression model that was developed fitted the data and explains 10.2% of the dependent variables. The statistically significant predictors for the Anxiety HADS-M dependent variables were: sex—female (*β* = 0.153; *p* = 0.009), the patients’ age (*β* = −0.177; *p* = 0.000), the NRS level of pain intensity (*β* = 0.201; *p* = 0.000) and not taking medication (*β* = −0.175; *p* = 0.036).

Younger age and lack of pharmacological treatment correlate with lower pain levels among the patients treated in the pain clinic. On the other hand, female sex and a higher NRS pain intensity has an impact on a higher level of anxiety among those examined.

Table 4 shows the regression analysis of the Depression HADS-M variable. The regression analysis turned out to fit but explained only 6.7% of the dependent variability. The statistically significant predictors turned out to be: sex—female (*β* = 0.224; *p* = 0.000), intensity of pain NRS (*β* = 0.222; *p* = 0.000) and the time of being treated in the pain clinic (*β* = 0.115; *p* = 0.026). Female sex, a higher level of pain intensity assessed on the NRS scale and a longer treatment time are associated with a higher level of depression among the patients.

The regression analysis carried out for the HADS-M: Aggression showed that the model developed was well-fitting and explains 6.7% of the dependent variability. Statistically significant predictors were: age (*β* = −0.221; *p* = 0.000) and pain intensity NRS (*β* = 0.116; *p* = 0.000). Analysis revealed that the younger the age of the patients the higher their level of aggression, whereas higher NRS pain intensity led to higher levels of aggression (Table 5).

## 4. Discussion

Pain is a multi-faceted and multi-aspect phenomenon, as well as a very unique personal experience. Perceived by patients as a disease and suffering, in the long term it leads to emotional problems, such as anxiety or depression, which makes it one of the dominant reasons for seeking help from healthcare specialists [2,3,4,19]. This is why the authors of the present paper made an attempt to specify the frequency of emotional disturbances in the form of anxiety, depression and aggression among pain clinic patients and the factors that have an impact on these disorders.

Our own research seems to have justified analyzing emotional reactions and the factors influencing their intensity, particularly that the study was conducted in pain clinics, which seem the best place for assessing biological, psychological and social factors among patients suffering from pain. The demographic features of the group examined, i.e., female domination, the 65–79 age bracket (mean age of 62.15) are comparable to those in studies by other authors: Ahmed et al. [4], Castro et al. [20], and Rockett et al. [21]. The results of our own research showed that over three fourths of the patients studied are under the continuous care of pain clinics. Only one fourth reported their first visit. The time of treatment usually ranged from 2 to 5 years. Osteoarticular and neuropathic pain dominated in the group examined. The research results of Rockett et al., who analyzed the types and character of pain among patients in a Plymouth pain clinic, show that back pain was diagnosed in 42% of those examined, musculoskeletal disorders in 30–40%, and neuropathic pain in 8–16% [21]. On the other hand, the research done by Brzeziński et al., which analyzed pain among those treated the Institute of Countryside Medicine in Lublin, pointed out that most of their patients suffered from spinal diseases and headaches [22]. Pain intensity was assessed as low in this group: 4.30 on the NRS scale.

Pain is closely connected with psychological problems, such as depression or anxiety, which moreover positively correlate with its intensity and other features. This is underlined by the literature on the subject [4,8,19,23,24]. Research done by Ahmed et al. on the frequency of anxiety and depression among patients of a pain clinic showed that depression ranked first, followed by anxiety [20]. Similar results were obtained by Bergander et al. [5], whereas the study by Sagheer et al. showed that chronic pain patients more often suffer from anxiety, followed by depression. [24]. This is confirmed by the research results of Soares-Filho et al. [25]. Our own research results show that the most frequent disorder among our patients was aggression, then anxiety and depression. This corresponded with the findings of Krzemińska et al. in the study on selected emotional problems suffered by the care-providers of patients fed by catering establishments in their home environments [16]. It must be pointed out that our research, similarly to the study by Krzemińska et al. [16], is broadened by the aspect of aggression, which is the third element on the HADS-M scale, modified to suit Polish conditions. Foreign research uses the original HADS scale [5,8,20,25].

Furthermore, the authors of the present paper assessed the factors determining anxiety, depression and aggression among pain clinic patients. The research of Elbinoune et al. on chronic neck pain and anxiety-depression problems shows that anxiety is significantly affected by such factors as education and the Visual Analogue Scale (VAS) score, whereas depression is influenced by education [26]. Zafar et al., on the other hand, did research on the manifold determinants of lower back pain, showing that it is affected by sex, age, the job performed, the patient’s working hours, co-morbidities, lifestyle—comprising e.g., sleeping problems, a soft mattress for sleeping, prolonged sitting or lifting heavy objects, as well as anxiety, stress or depression [27]. Fernandes et al. conducted a cross-sectional study of neuropathic knee pain and its risk factors in Great Britain. The results of their study showed that the condition they studied is more often diagnosed e.g., in younger people, women, people with a higher BMI, patients prone to self-diagnosing fibromyalgia, hypertension, hyperlipidemia and diabetes, those who had suffered a knee injury, do high-risk jobs, suffer from pain in general, and have higher results for anxiety and depression than do people without knee pain [28]. Da Luz et al., in turn, in their research on the living conditions of women with chronic pelvic pain showed that the factors that have a negative impact on the studied women’s quality of life were: anxiety, depression, sexual disfunction, hypertension, diabetes, pain intensity, lower income in the family and the lack of a partner [29]. Our own study showed, on the other hand, that among the patients of pain clinics a higher level of anxiety was ascertained in women and people with a higher pain intensity (NRS). Younger age and the lack of pharmacological treatment correlated with a lower level of anxiety. A higher level of depression was diagnosed among women, in cases of higher pain intensity (based on the NRS scale) and a longer time of treatment.

The results presented above regarding the analysis of anxiety and depression based on the HADS-M scale also include a third scale, i.e., aggression [15,16]. Aggression is a psycho-behavioral symptom defined as—overt, often harmful social interaction with the intention of inflicting damage or other unpleasantness upon another individual often accompanied by physical pain [30]. Krzemińska et al. found in their study based on the HADS-M scale that in the group of care-givers, fully aggressive behavior was more often displayed in women than in men [16]. Park and Seo in their study on aggression and its correlation with suicide among patients with migraines concluded that the factors impacting the research results on the basis of the Aggression Questionnaire were: anxiety, the intensity of the headache and chronic migraine [30]. Bergander et al. on the other hand, conducted research analyzing younger and older patients with chronic somatic pain from the point of view of psychodiagnostics, relations between the doctor and the patient, and treatment results. They showed that age was a significant factor in terms of the level of aggression, i.e., younger patients were likely to be more aggressive than older ones [5]. Our own research results confirm the above statement, since a higher level of aggression was connected with younger age. In addition, greater aggression was connected with higher pain intensity (NRS).

The authors of the present paper made an attempt to present the conditions impacting anxiety, depression and aggression among patients of pain clinics, as they have not found such studies in the literature so far. The HADS—M scale as a screening tool that evaluates behavior in a population of patients with pain and somatic symptoms makes it possible to classify emotional behaviors. Not one symptom but a set of specific behaviors ranked as a scale helps the doctor diagnose a specific disorder—anxiety, depression or aggression.

Nevertheless, our research has some limitations. The study was conducted in only two pain clinics in Poland, focused on only a few types of pain and it analyzes selected determinants.

Social awareness of the co-occurrence of anxiety, depression and aggression with somatic diseases is low. Routine use of the questionnaire in pain counseling centers may contribute to reaching early diagnoses, classifying the emotional disorders that accompany pain syndromes and planning therapy, not only to ease the patients’ discomfort but help them return to active family and social life. As the patients’ clinical condition improves, their disability should diminish, thus reducing the involvement of third parties in caregiving and physiotherapy. Since the scale is simple and easy to use, it can benefit not only the patients of basic health care and specialist clinics catering for somatically ill people but also those who are healthy.

The results discussed in this paper and their analysis will help improve care for pain clinic patients, develop better treatment methods, diminish adverse side effects and increase patient satisfaction. Therapies including medications that relieve emotional disorders will make it possible to intensify pain management methods and develop multimodal therapy.

## 5. Conclusions

The most frequent emotional disturbances among the patients examined were aggression, anxiety and depression, in this order. The factors determining their anxiety level were: sex, age, pain intensity and the time of being treated in the clinic, whereas the level of depression was influenced by age and pain intensity.

## Figures and Tables

**Table 1 ijerph-15-02898-t001:** Characteristics of the patients examined.

**Sex**	**n (%)**
Female	653 (63.71)
Male	372 (36.29)
**Age**	**n (%)**
<34 years	60 (5.85)
35–49 years	151 (14.73)
50–64 years	328 (32.00)
65–79 years	361 (35.22)
≥80 years	125 (12.20)
Age (years)—mean (SD)	62.15 (15.48)
**Type of pain**	**n (%)**
Osteoarticular	447 (43.61)
Neuropathic	426 (41.56)
Headache	120 (11.71)
Others	104 (10.15)
**Pharmacological treatment**	**n (%)**
Yes	557 (54.34)
No	468 (45.66)
**Medication taken**	**n (%)**
Antidepressants	493 (88.51)
Benzodiazepines	20 (3.59)
Sleeping pills	47 (8.44)
**Visit in the clinic**	**n (%)**
First time	256 (24.98)
Subsequent	769 (75.02)
**Time of treatment in the clinic**	**n (%)**
Up to 3 months	134 (17.43)
4–6 months	104 (13.52)
7–12 months	110 (14.30)
13–23 months	131 (17.04)
2–5 years	188 (24.45)
Over 5 years	102 (13.26)
NRS scale—mean (SD)	4.30 (2.87)

**Table 2 ijerph-15-02898-t002:** Assessment of anxiety, depression and aggression among the patients examined (HADS-M scale).

HADS-M Scale	
HADS-M ANXIETY	n (%)
Lack of disturbances	419 (40.88)
Borderline cases	274 (26.73)
Disturbances occur	332 (32.39)
HADS-M ANXIETY—mean (SD)	8.67 (4.14)
HADS-M DEPRESSION	n (%)
Lack of disturbances	584 (56.98)
Borderline cases	258 (25.17)
Disturbances occur	183 (17.85)
HADS-M DEPRESSION—mean (SD)	7.00 (4.02)
HADS-M AGGRESSION	n (%)
Lack of disturbances	388 (37.85)
Borderline cases	164 (16.00)
Disturbances occur	473 (46.15)
HADS-M AGGRESSION—mean (SD)	3.24 (1.72)

**Table 3 ijerph-15-02898-t003:** Regression Analysis for the HADS-M: Anxiety.

Selected Predictors	HADS-M Anxiety *R*^2^ = 0.102 F = 8.850 *p* < 0.001			
	*B*	*Β*	*T*	*p*
Sex	1.467	0.153	2.627	0.009
Age	−0.047	−0.177	−5.859	0.000
NRS	0.290	0.201	6.539	0.000
Osteoarticular pain	−0.372	−0.045	−0.746	0.456
Neuropathic pain	−0.193	−0.023	−0.394	0.694
Headache	0.107	0.008	0.193	0.847
Other types of pain	−0.037	−0.003	−0.059	0.953
Taking benzodiazepine	−0.265	−0.009	−0.279	0.781
Taking sleeping pills	0.410	0.020	0.584	0.559
Lack of pharmacological treatment	−1.457	−0.175	−2.095	0.036
Time of treatment	0.001	0.047	0.938	0.348

**Table 4 ijerph-15-02898-t004:** Regression Analysis for the HADS-M: Depression.

Selected Predictors	HADS-M: Depression *R*^2^ = 0.067 F = 5.582 *p* < 0.001			
	*B*	*Β*	*T*	*p*
Sex	2.084	0.224	3.767	0.000
Age	0.001	0.003	0.109	0.913
NRS	0.312	0.222	7.109	0.000
Osteoarticular pain	−0.510	−0.063	−1.032	0.302
Neuropathic pain	−0.268	−0.033	−0.550	0.582
Headache	−0.269	−0.022	−0.491	0.624
Other types of pain	−0.005	0.000	−0.008	0.994
Taking benzodiazepine	−1.417	−0.049	−1.505	0.133
Taking sleeping pills	0.468	0.024	0.674	0.501
Lack of pharmacological treatment	−1.825	−0.226	−2.650	0.008
Time of treatment	0.002	0.115	2.234	0.026

**Table 5 ijerph-15-02898-t005:** Regression analysis for the HADS-M: Aggression.

Selected Predictors	HADS-M Aggression *R*^2^ = 0.067 F = 5.600 *p* < 0.001			
	*B*	*β*	*T*	*P*
Sex	−0.063	−0.016	−0.268	0.788
Age	−0.025	−0.221	−7.164	0.000
NRS	0.069	0.116	3.696	0.000
Osteoarticular pain	−0.032	−0.009	−0.151	0.880
Neuropathic pain	−0.066	−0.019	−0.318	0.750
Headache	0.071	0.013	0.303	0.762
Other types of pain	−0.056	−0.010	−0.213	0.831
Taking benzodiazepine	−0.294	−0.024	−0.729	0.466
Taking sleeping pills	0.183	0.022	0.616	0.538
Lack of pharmacological treatment	−0.287	−0.083	−0.975	0.330
Time of treatment	0.000	−0.030	−0.576	0.564

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
