# Peer review of "Analysis of Anxiety, Depression and Aggression in Patients Attending Pain Clinics"

_ijerph, 2018, doi:10.3390/ijerph15122898_

Round 1

Reviewer 1 Report

This is a noteworthy paper, which aim was set as its objective to measure the frequency of emotional disturbances in the form of anxiety, depression and aggression among patients treated in pain clinics and to specify the factors contributing to the conditions that influence them.

The main strengths  of the paper is large sample size - 1025 people.

The article should be published and the readers can benefit from the facts presented in the paper.

I have some minor comments regarding the article:

Introduction

In the part introduction the authors described only the problem of pain. I suggest adding a few introductory sentences to the problem of pain, anxiety and aggression.

Materials and Methods

Participants  - Page 2, line 70: 

The exclusion criteria, on the other hand, were: lack of informed consent on the part of the patient, and lack of ascertained pain symptoms - I am not sure if the second exclusion criterion needs to be placed.

The psychometric properties of the HADS-M scale should be added.

Assessment – Page 3., line 96 - reference 16 should be removed.

Table 1. Characteristics of the patients examined – part time of treatment in the clinic. The following ranges were used in the work: up to 3 months; 3-6 months, 6-12 months e.t.c.

The analyzed time intervals coincide, it should be: up to 3 months; 4-6 months e.t.c.

Discussion

Maybe the authors could elaborate a bit more on what these results could mean to the health care providers?

Author Response

Dear Reviewer,

The authors would like to thank the Reviewer for many valid comments and suggestions. We have revised our paper as requested. Please find a point-by-point response to the reviewer’s comments below.

The paper has been proofread again by a native speaker of English in a professional translation agency.

This is a noteworthy paper, which aim was set as its objective to measure the frequency of emotional disturbances in the form of anxiety, depression and aggression among patients treated in pain clinics and to specify the factors contributing to the conditions that influence them.

The main strengths of the paper is large sample size - 1025 people.

The article should be published, and the readers can benefit from the facts presented in the paper.

I have some minor comments regarding the article:

Introduction

In the part introduction the authors described only the problem of pain. I suggest adding a few introductory sentences to the problem of pain, anxiety and aggression.

We are grateful for your important observation. The missing information was added to the introduction following your suggestions.

Materials and Methods

Participants - Page 2, line 70: 

The exclusion criteria, on the other hand, were: lack of informed consent on the part of the patient, and lack of ascertained pain symptoms - I am not sure if the second exclusion criterion needs to be placed.

Changes were made in accordance with the suggestion – the second exclusion criterion was removed.

The psychometric properties of the HADS-M scale should be added.

The psychometric properties of the HADS-M scale were added in accordance with the reviewer’s suggestion.

Assessment – Page 3., line 96 - reference 16 should be removed.

The authors think that reference 16 should remain because it includes important elements regarding the research tool. Moreover, the results of the research presented there have been used in the discussion.

Table 1. Characteristics of the patients examined – part time of treatment in the clinic. The following ranges were used in the work: up to 3 months; 3-6 months, 6-12 months e.t.c.

The analyzed time intervals coincide, it should be: up to 3 months; 4-6 months e.t.c.

Thank you for your observation. The editorial error in Table 1 has been corrected.

Discussion

Maybe the authors could elaborate a bit more on what these results could mean to the health care providers?

Thank you for your suggestion. Considerations on the research results for health service employees and tips on how to use them in practice were added.

All the changes have been highlighted in yellow.  

Reviewer 2 Report

Overall the paper appears to have merit. It is of interest and seems to address a relevant clinical issue. It is well-organized. This reviewer sees the following issues to be addressed upon revision:

The manuscript needs a through editing for preciseness and brevity. Some examples:

 Line 59. Change its to their

Line 110. Change variant to variable

Line 129-130. Edit overly wordy statements such as In the studied group of patients…. to Anxiety was diagnosed in one third of the patients …

Also

Table 2. Bottom Line. What is AGRESJA?

Line 85. Is there a citation for these authors?

Three critical issues that must be addressed:

The authors employ a questionnaire type assessment to determine the incidence of two specific DSM-V defined clinical entities: anxiety and depression. How reliably does their questionnaire identify individuals who are actually diagnosed or diagnosable with these conditions? As for aggression, (and this is nicely defined in the Discussion) this is also an outward manifestation of hostility and/or paranoia, i.e., a sign of a clinically diagnosable mood disorder. So the same questions hold for aggression.

Granted, one could also argue that whether or not their questionnaire has validity for diagnosing these clinical conditions, the important issue is that it have validity for improving their clinical care. This should be addressed in the Discussion.

Along these lines the descriptors used in the tables such as “Lack of disturbances, Borderline cases, and Disturbances occur” seem immensely vague.  Could these phrases be related more precisely to either a clinical diagnosis or specific behaviors which are problematic in the pain clinic or to the patient’s overall functioning with caregivers?

Finally, for the Discussion, having developed these somewhat elaborate statistical models to identify their patient’s profiles and issues, what will the authors do with this information now that they have it? How will it improve patient care?

The Discussion could be shortened by one third with editing for conciseness.

Author Response

Dear Reviewer,

The Authors would like to thank the Reviewer for many valid comments and suggestions. We have revised our paper as requested. Below please find our point-by-point response to the reviewer’s comments.

The paper has been proofread again by a native speaker of English in a professional translation agency.

Overall the paper appears to have merit. It is of interest and seems to address a relevant clinical issue. It is well-organized. This reviewer sees the following issues to be addressed upon revision:

The manuscript needs a through editing for preciseness and brevity. Some examples:

 Line 59. Change its to their

The change was made according to the native speaker suggestions.

Line 110. Change variant to variable

Line 129-130. Edit overly wordy statements such as In the studied group of patients…. to Anxiety was diagnosed in one third of the patients …

Changes were made according to the suggestions.

Also

Table 2. Bottom Line. What is AGRESJA?

This was an omission in the translation. The change was made to Aggression.

Line 85. Is there a citation for these authors?

Yes. This citation is at the end of the paragraph.

Three critical issues that must be addressed:

The authors employ a questionnaire type assessment to determine the incidence of two specific DSM-V defined clinical entities: anxiety and depression. How reliably does their questionnaire identify individuals who are actually diagnosed or diagnosable with these conditions? As for aggression, (and this is nicely defined in the Discussion) this is also an outward manifestation of hostility and/or paranoia, i.e., a sign of a clinically diagnosable mood disorder. So the same questions hold for aggression.

The Hospital Anxiety and Depression Scale (HADS) is a commonly used method of measuring the level of anxiety and depression both in psychiatric practice and in studies of mentally healthy people whose emotional status should for some reason be assessed. Two statements assessing the level of aggression have been added to the classical HADS scale because disease and pain can be connected not only with anxiety and depression but also with rage, irritability and anger. Validation studies of the basic and modified version of the HADS scale showed its high reliability and accuracy. This scale, as a screening tool evaluating behavior in a population of patients with pain and somatic symptoms, makes it possible to classify these patients in a specific group of emotional behaviors (anxiety, depression and aggression). It is not only a symptom but a set of specific behaviors ranked in a scale that allows one to identify a specific disorder - anxiety, depression or aggression. Interpretation of individual behaviors or emotional symptoms in isolation from the scale may lead to erroneous statements and unauthorized conclusions because similar symptoms may occur in various mental disorders.

Granted, one could also argue that whether or not their questionnaire has validity for diagnosing these clinical conditions, the important issue is that it have validity for improving their clinical care. This should be addressed in the Discussion.

Thank you for your suggestion. The missing aspect has been included in the Discussion.

Along these lines the descriptors used in the tables such as “Lack of disturbances, Borderline cases, and Disturbances occur” seem immensely vague.  Could these phrases be related more precisely to either a clinical diagnosis or specific behaviors which are problematic in the pain clinic or to the patient’s overall functioning with caregivers?

The phrases used in table 2 refer to the entire population of patients analyzed in the study identifying groups with emotional disorders. The relationship and correlation of mental disturbances with somatic disorders, treatment methods, demographic data and the type of pain are included in Tables 3, 4, 5.

Finally, for the Discussion, having developed these somewhat elaborate statistical models to identify their patient’s profiles and issues, what will the authors do with this information now that they have it? How will it improve patient care?

Thank you for your valuable observation. As suggested, the importance of the results obtained for patients and the way they should be implemented in the treatment of pain has been added to the Discussion.

The Discussion could be shortened by one third with editing for conciseness.

The authors are grateful for the suggestion but we have made every effort to make the discussion as thorough and as accessible to the reader as possible.

All the remarks and suggestions addressed in the text have been highlighted in yellow.  
